# The low and high doses administration of lutein improves memory and synaptic plasticity impairment through different mechanisms in a rat model of vascular dementia

Hamideh Asadi nejad[1], Amirhossein Yousefi Nejad[2], Somayeh Akbari[3], Maryam Naseh[3], Seyed Mostafa Shid Moosavi[1], Masoud Haghani [1,3]*

1 Department of Physiology, The Medical School, Shiraz University of Medical Sciences, Shiraz, Iran,
2 Faculty of Veterinary Medicine, Department of Veterinary Medicine Islamic Azad University of Kazeroon, Shiraz, Iran, 3 Histomorphometry and Stereology Research Center, Shiraz University of Medical Sciences, Shiraz, Iran

* haghani@sums.ac.ir

## Abstract

### Background and aim

Vascular dementia (VD) is a common type of dementia. This study aimed to evaluate the effects of low and high doses of lutein administration in bilateral-carotid vessel occlusion (2VO) rats.

### Experimental procedure

The rats were divided into the following groups: the control, sham-, vehicle (2VO+V) groups, and two groups after 2VO were treated with lutein 0.5 (2VO+LUT-o.5) and 5mg/kg (2VO +LUT-5). The passive-avoidance and Morris water maze were performed to examine fear and spatial memory. The field-potential recording was used to investigate the properties of basal synaptic transmission (BST), paired-pulse ratio (PPR), as an index for measurement of neurotransmitter release, and long-term potentiation (LTP). The hippocampus was removed to evaluate hippocampal cells, volume, and MDA level.

### Result

Treatment with low and high doses improves spatial memory and LTP impairment in VD rats, but only the high dose restores the fear memory, hippocampal cell loss, and volume and MDA level. Interestingly, low-dose, but not high-dose, increased PPR. However, BST recovered only in the high-dose treated group.

### Conclusions

Treatment with a low dose might affect neurotransmitter release probability, but a high dose affects postsynaptic processes. It seems likely that low and high doses improve memory and LTP through different mechanisms.

s

**Data Availability Statement:** Data cannot be shared publicly. Data are available from the Shiraz medical university, (contact via haghani@sums.ac. ir) for researchers who meet the criteria for access to confidential data. To ensure long-term stability and availability of the data, we established a data access point within our institution. We provided contact details, including phone/email, for our data access committee with following addresses: Mrs Masoumi, phone number +987132302026 E-mail: medphyzio1@sums.ac.ir The data access committee within our department comprises several members, including Mrs. Masoumi, who serves as the chairperson of the committee. As such, Mrs. Masoumi is chairperson of Data Access Committee of physiology department.

**Funding:** The Research Council of Shiraz University of Medical Sciences, Shiraz, Iran, financially supported this study (Grant Number: 26316), as a thesis of Mrs. Asadi nejad for acquiring M.Sc. degree in physiology. The funders had no role in study design, data collection and analysis, decision to publish, or preparation of the manuscript.

**Competing interests:** The authors have declared that no competing interests exist.

## 1. Introduction

Chronic cerebral hypoperfusion (CCH) induces memory impairment and cognitive dysfunction and may lead to vascular dementia (VD). The decrease of blood flow to the brain areas that are responsible for cognitive functions, especially the hippocampus, leads to the suffering of the neurons for a long period from nutrients, energy, and oxygen [1]. Thereby, the hypoperfusion could be associated with a decrease in glucose metabolism and an increase in oxidative stress and neuroinflammation. The neuroinflammation and oxidative stress are generally seen as factors that are strongly related to disruption of the blood-brain barrier [6], neural circuits [2], neurotransmitter synthesis and release [3], and impairment of synaptic plasticity [4]. All these factors can impact cell survival which leads to a progressive loss in cognitive functions, as observed in VD. It is well known that long-term potentiation (LTP) is a cellular base for memory storage. Much of the literature emphasizes that neurodegenerative disorders such as AD and VD are associated with cognitive loss and exhibit synaptic plasticity impairment. Indeed, it is a useful tool to evaluate, presynaptic release function, and synaptic plasticity to assess pathophysiology and treatment approaches in neurodegenerative disease.

Although there is no effective treatment for CCH-induced vascular dementia, several studies have reported beneficial effects of various compounds [5, 6], recently our group has also shown neuroprotection of platelet-rich plasma [7], edaravone [8], and epidermal neural crest stem cells [9] in VD rats with different molecular and cellular therapeutic effects. According to these data, targeting oxidative/nitrosative stress and neuroinflammation could be a promising therapeutic strategy for VD treatment. Oxidative stress occurs upon excessive free radical formation and impairs neurotransmitter transmission, the integrity of the plasma membrane, and whole brain neural activity, and leads to neuronal apoptosis [10]. Antioxidants are compounds in foods that scavenge and neutralize free radicals and ROS that have been implicated in the pathology of several neurodegenerative diseases [11]. Therefore, intake of antioxidants may counteract with harmful effects of free radicals on cognitive function. Previous research has established that a decrease in plasma antioxidants, such as lutein and zeaxanthin is tied to cognitive impairment in AD patients [12, 13].

Lutein, as an antioxidant, ameliorates oxidative damage against free radicals [14, 15]. Lutein and zeaxanthin, are two carotenoids that are present in high concentrations in the retina and brain but cannot be synthesized in the body, they must be supplied in the diet [15]. More interestingly, lutein is the most predominant carotenoid in the human brain tissue in early and late life, especially, it is presented in the areas that responsible for the control of cognitive function [16]. A great deal of previous research provides further evidence that cognitive functions are related to lutein levels in adult brain tissue [13]. Previous research has established that lutein has an essential neuroprotection in the eyes and brain against retinal detachment, cerebral ischemia, Huntington's disease, Parkinson's disease, and Alzheimer's disease [17–21]. Surprisingly, the effects of lutein have not been closely examined, therefore, there are two primary aims: 1. To investigate the usefulness of lutein in VD and 2. To determine the potential cellular mechanism of lutein. With these aims, we evaluated spatial memory and synaptic plasticity by Morris water maze (MWM) test and field potential recording respectively. We also measured the MDA level and the cell number and volume of hippocampal CA1.

## 2. Materials and methods

### 2.1. Animals

Adult male Sprague Dawley rats (250–300 g, aged 7–10 weeks) were housed under a T = 24 cycle (12 h light, 12 h dark) with ad libitum access to food and tap water in a temperature of

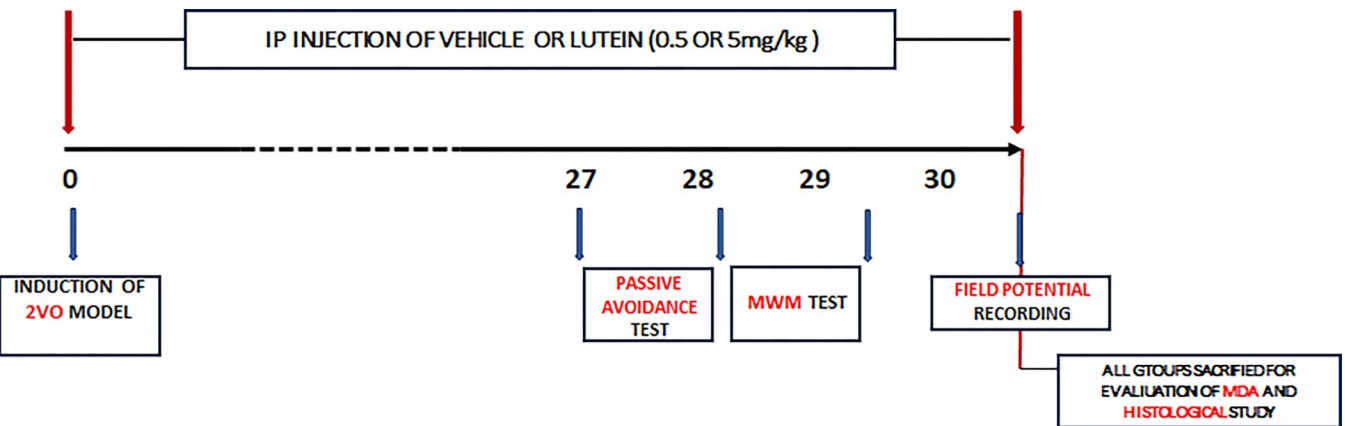

**Fig 1. The schematic representation of the protocol, the timeline for passive avoidance, Moris water maze (MWM) tests, and field potential recording in 2-vessel occlusion (2VO) rats.**

about 22˚C and proper air humidity. The animal studies were carried out in compliance with the approved protocols and guidelines of the Institutional Ethics Committee of Shiraz University of Medical Sciences (IR.SUMS.AEC.1401.077). To alleviate animal suffering, all animals euthanized at end of the experiment, following induction of deep anesthesia with urethane, rats were sacrificed using a guillotine. The personnel responsible for sacrifice were trained in the humane and ethical handling of animals, and the procedures were conducted in compliance with the Institutional Ethics Committee of Shiraz University of Medical Sciences.

The rats were randomly assigned to five groups as follows: the control (n = 8), sham-operation (sham, n = 7), the vehicle group, after bilateral common carotid occlusion, was treated with almond oil as vehicle (2VO +V; n = 7), two groups after 2VO, were treated by lutein (Danna pharma Co.) 0.5 and 5mg/kg (2VO+LUT-o.5; n = 9 and 2VO+LUT-5; n = 8, respectively). The treated rats received lutein or vehicle immediately after 2VO, and the treatment continued once daily for 30 days (Fig 1).

## 2.2. Bilateral common carotid arteries occlusion

The CCH was induced by 2VO as described previously in more detail [7, 22]; briefly, the rats were anesthetized with ketamine (90 mg/kg) and xylazine (2 mg/ kg) and placed in the supine position on the operating table. Incisions were made in the skin to expose and ligate the right and left common carotid arteries using 5–0 surgical silk. The sham operation group underwent an identical surgical procedure without ligation of the common carotid arteries. (Fig 1).

## 2.3 Behavioral study

**2.3.1 Passive avoidance test.** A shuttle box apparatus with two black and white chambers separated by Plexiglas was used to assess the passive avoidance test. The rats have an innate preference for the dark compartment. On the 27th day post-surgery, the suppression of the innate preference was done by placing the animal in the white chamber, and the animals were punished upon entrance to the dark compartment by an electrical foot shock (0.5 mA, 50 Hz, 2 s once). After the learning acquisition, the rats avoid going to the black side. One day later, the test was repeated without foot shock, and the delay time before the entrance to the black chamber was recorded as step-through latency (STL) [23].

**2.3.2 Morris Water Maze (MWM).** The MWM test was used to evaluate hippocampal spatial learning and memory. The maze consists of a large black circular swimming pool with

an escape platform in the target quadrant. On day 29 post-surgery, the rats were trained to find a visible escape platform which was placed 1.5 cm above the surface water. The learning was evaluated while the platform was submerged 1.5 cm below the surface water. The time spent to find the hidden platform was recorded for each animal. We evaluated the progress of learning via three blocks and each block contains 4 trials. Twenty-four hours after the last trial (On day 30 post-surgery), the platform was removed to evaluate spatial memory tests (probe test). The percentage of time spent in the target quadrant was recorded as an index for spatial memory retention [24].

## 2.4 Electrophysiological study

On day 30 post-surgery the synaptic transmission and plasticity at CA3-CA1 were assessed using field potential recordings as previously described in more detail [25]. Each animal was anesthetized by intraperitoneal injection of urethane (1.5 gr/Kg) and placed in the stereotaxic frame. The Schaffer collateral pathway lies between the CA3 and CA1 regions of the hippocampus. The stimulating electrode was located on the stratum radiatum of the CA1 region to stimulate the Shaffer collateral pathway and the recording electrode was placed in the CA1 area through the small holes that were drilled in these areas. The evoked field excitatory postsynaptic potential (fEPSP) was recorded after 25 minutes of rest. The basal synaptic transmission (BST) was evaluated by the input/output I/O curve. The high-frequency stimulation (HFS) was delivered to induce LTP. The level of LTP induction was calculated as the percentage change of fEPSP amplitude after HFS delivery compared to the mean amplitude before HFS [26, 27].

## 2.5. Stereological study

The hippocampal CA1 volume and number of neurons were calculated by stereological technique, as previously described in more detail [28]. Firstly, the hemispheres were fixed for 10 days in neutral buffered formaldehyde, the fixed tissue underwent tissue processing and then was embedded in paraffin, and the paraffin block (25 μM) was sectioned and stained by Giemsa (Merck, Germany). Finally, the volume of the CA1 region was calculated by using Cavalieri's principle, and the total pyramidal cell number in the CA1 region was assessed by the optical dissector method [8].

## 2.6. Malondialdehyde (MDA) assay

At the end of the recording, the hippocampus was removed and the thiobarbituric acid-reactive substances (TBARS) test was used for measuring MDA levels as described previously [8]. Briefly, the samples were homogenized and centrifuged at 12,000 rpm for 15 minutes. The supernatants and standards (1,1,3,3-tetra ethoxy propane) were mixed with a solution containing 0.25 N hydrochloric acid (HCl), 20% trichloroacetic acid (TCA), and 0.8% thiobarbituric acid (TBA). The mixture was then incubated at 87°C for 1 hour and centrifuged at 12,000 rpm for 5 minutes. Finally, the absorbance wavelength of the samples was read using a microplate reader at 532 nm.

## 2.7. Statistical analysis

The data are presented as Mean ± SEM. One-way ANOVA with Tukey's post-hoc test was used to compare STL, swimming speed, percentage of time spent in the correct quadrant of the MWM, paired-pulse ratio, and EPSP amplitudes in the input/output (I/O) curve, MDA level, cell number, and cell volume. The mean amplitude of EPSPs before and after HFS and

the levels of paired-pulse ratio (PPR) changes before and after HFS delivery were compared by paired t-test. Two-way repeated-measures ANOVA was used to compare the time spent finding the platform and the fEPSP percent changes after delivery of HFS. All data analyses were performed using PRISM.9 software and statistical significance was considered at P < 0.05.

## 3. Results

### 3.1. Behavioral experiments

**3.1.1. High but not low doses of lutein improve passive avoidance memory in 2VO rats.** Fig 2 compares the STL time of the passive avoidance test. The 2VO was found to lead to a decrease in STL time compared to the sham group (55.5±13.82s *vs* 239.3±26.41s; *P<0.001*). The administration of low-dose lutein (0.5mg/kg) in the 2VO+LUT-0.5 group did not affect on STL time (114.9±32.04s). However, the high-dose lutein (5mg/kg) increased the STL time (165.9±30.73.47s) to the level that was comparable respect to the sham, but significantly (*P<0.05)* higher compare to the low-dose.

**3.1.2. Low and high dose of lutein improves spatial learning and memory in 2VO rats.** The results of the MWM test showed that during the training blocks (B1, B2 and B3), the 2VO was found to impaired significantly (*P<0.05)* learning levels and the rats in this group spent more time for finding hidden platform (B1 = 49.85±4.03s; B2 = 42.92±4.35s and B3 = 44.48 ±4.28s) respect to sham (B1 = 36.85±3.61s; B2 = 26.84±4.16s and B3 = 28.5±4.11s) and treated groups (*P<0.05- P<0.001)*. However, both the low-dose (B1 = 35.98±3.96s; B2 = 26.13±4.09s

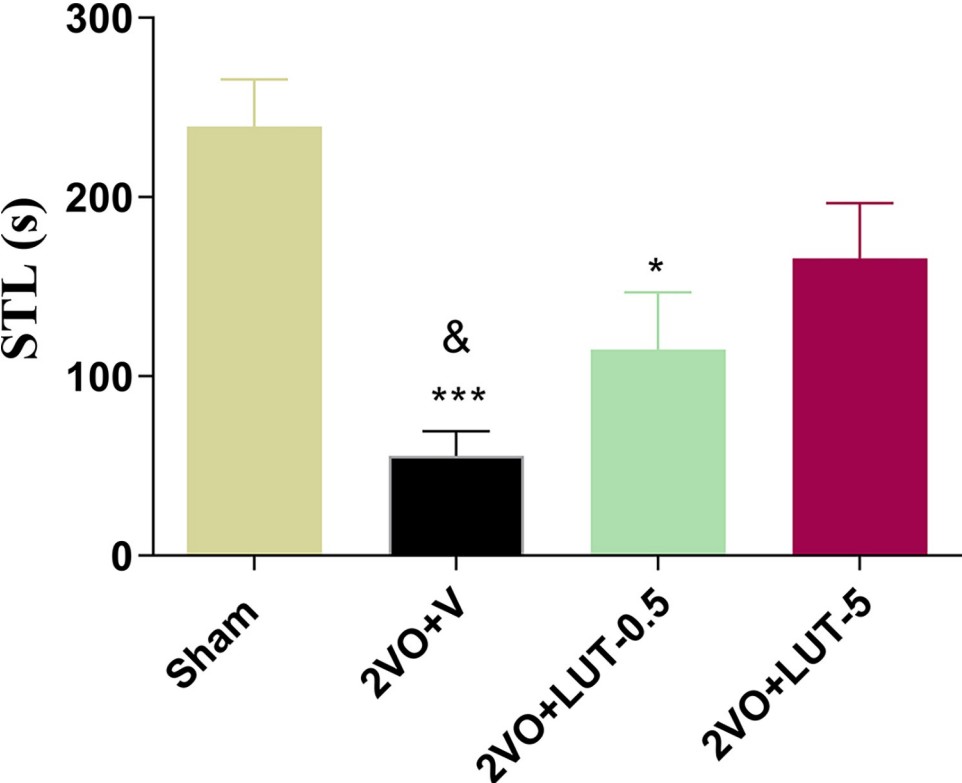

**Fig 2. The fear memory assessment by passive avoidance test.** High (5mg/kg) but not low (0.5mg/kg) doses of lutein improve passive avoidance memory in 2-vessel occlusion (2VO) rats. The values are shown as mean ± SEM. Significant differences with respect to the sham (*$P < 0.05$ and ***$P < 0.001$) and 2VO + LUT-5 (& $P < 0.05$). Sham (n = 7), 2VO + V (n = 7), 2VO + LUT-0.5 (n = 11) and 2VO + LUT-5 (n = 8).

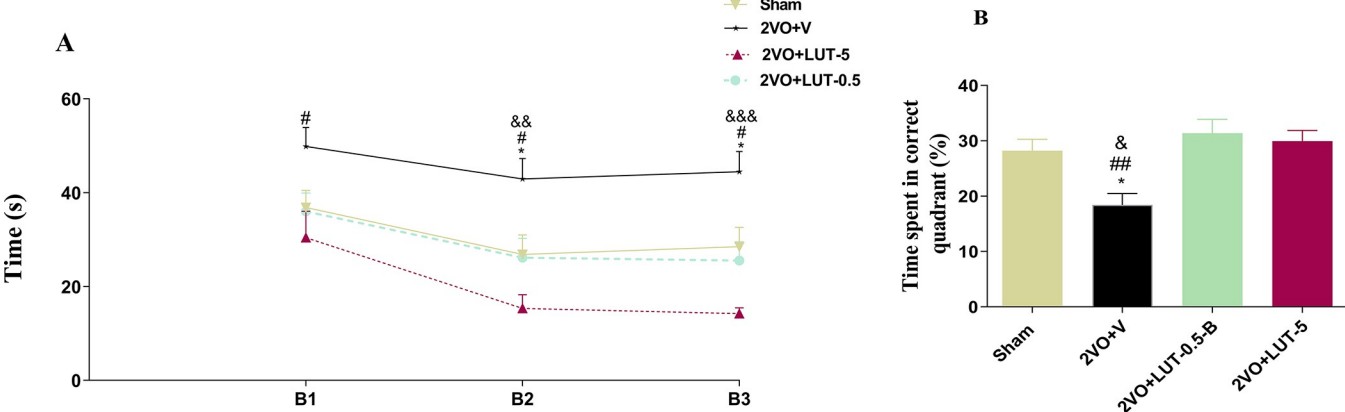

**Fig 3. The evaluation of spatial learning and memory by Morris water maze test.** Both low (0.5mg/kg) and high (5mg/kg) doses of lutein improve spatial learning and memory in (2-vessel occlusion) 2VO rats. The escape latency time during the three blocks in the 2VO + LUT-0.5 and 2VO + LUT-5 groups significantly decreased compared to the 2VO + V group (A). In the probe test, the rats from low and high-dose treated groups spent more time in the corrected quadrant relative to the 2VO + V group (B). (One-way ANOVA with post-hoc test). Significant differences with respect to the sham (*$P < 0.05$), 2VO + LUT-0.5 (#$P < 0.05$ and ## $P < 0.01$) and 2VO + LUT-5 ((& $P < 0.05$, && $P < 0.01$ and &&& $P < 0.001$). The values are shown as mean ± SEM. Sham (n = 7), 2VO + V (n = 7), 2VO + LUT-0.5 (n = 11) and 2VO + LUT-5 (n = 8).

and B3 = 25.53±3.98s) and high-dose (B1 = 30.42±5.6s; B2 = 15.35±2.91s and B3 = 14.23 ±1.22) groups as well as the sham group demonstrated significant trial effects in the learning procedure. It is interesting to note that no significant difference was found between the treated groups and the sham group (Fig 3A).

Twenty-four hours following the learning trials, the platform was removed for a probe trial to assess spatial memory retention, measured by the percentage of time spent in the target quadrant. The mean time spent for the 2VO+V group was18.41±2.08% which was significantly lowered ($P < 0.05$) relative to the sham group (28.23±2.08%) (Fig 3B). But, the low and high doses of lutein significantly restored the memory retention and time spent in the target quadrant (31.38±2.49% and 29.97±1.91% respectively) compare to the 2VO+V group. Since the swimming speed was not different among the groups, the latency to find the platform was used as an indicator of learning performance.

## 3.2. Electrophysiological results

**3.2.1. The administration of lutein did not affect impaired basal synaptic excitability of the CA1 neurons in 2VO rats.** The input-output curves were drawn by recording the amplitude of fEPSPs in CA1 following stimulation of Schaffer-collaterals with different stimuli strengths that started from 50 μA and continued until the maximum response. The result indicated a depression on the I/O curve of the 2VO+V group with respect to that of the sham group. This depression was also seen in both low and high-dose lutein-received groups (Fig 4A). For a better picture, the maximal fEPSP amplitude was evaluated, and the 2VO+V group was found to have a significantly decreased respect to the sham group (480.3±40.22 *vs.* 819.1± 74.82; $P < 0.05$). However, treatment with low and high doses of lutein failed to completely recover the maximal fEPSP amplitude (499.0 ± 68.31 *vs*. 548.4 ± 83.21; respectively) (Fig 4B). Although this parameter significantly lowered in 2VO+LUT-0.5 respect to the sham group, there were no significant differences between the sham and 2VO+LUT-5 groups, high dose treated rats also did not show significant differences respect to low dose and the 2VO+V groups.

**3.2.2. The low dose but not high dose of lutein increased short-term synaptic plasticity in 2VO rats.** The short-term plasticity was evaluated by the paired-pulse ratio (PPR) in inter-

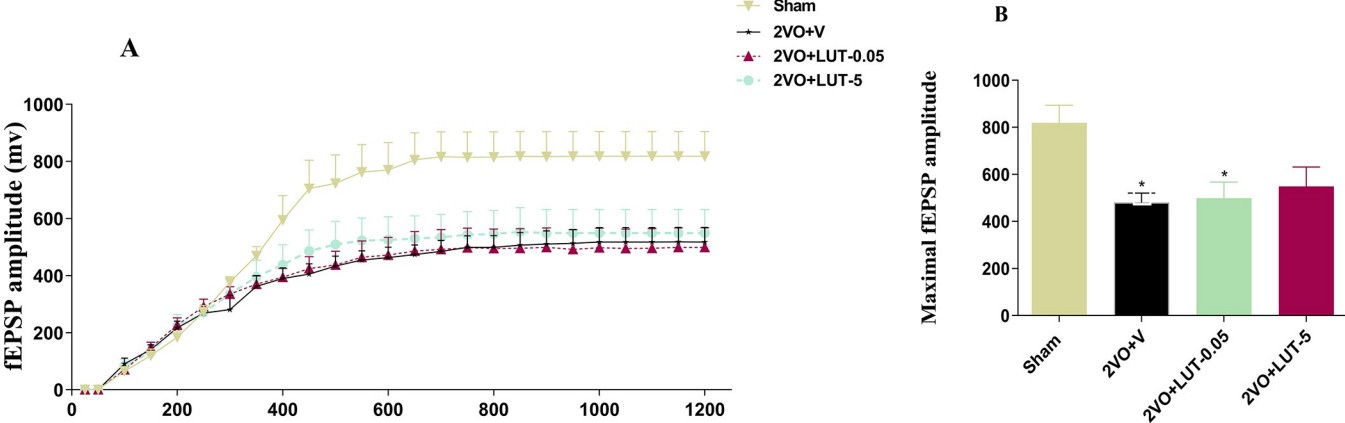

**Fig 4.** The basal synaptic transmission (BST) was evaluated by input/output curve (A). The administration of a low dose (LUT-0.5) of lutein did not affect impaired BST of the CA1 neurons in 2-vessel occlusion 2VO rats, but we found the partial recovery of BST only in the high dose (5mg/kg) lutein-treated group. A significant decrease in the half-maximal fEPSP amplitude of 2VO + V and 2VO + LUT-0.5 groups relative to the sham group, there were no significant differences between the sham and 2VO+LUT-5 groups (B). Significant differences with respect to the sham (*$P < 0.05$). The values are shown as mean ± SEM. Sham (n = 7), 2VO + V (n = 7), 2VO + LUT-0.5 (n = 11) and 2VO + LUT-5 (n = 8).

stimulus intervals (ISI) from 50 to 250 ms (Fig 5A). In Fig 5B there is a clear trend of decreasing in PPR from ISI 50 to 250 in all groups. What stands out in this figure is the higher levels of PPR in the 2VO+Lut-0.05 group with respect to all groups. Although, the result did not show statistical differences compared to the sham and 2VO+Lut-5, interestingly from ISI 50 to 175 the PPR values of low dose group were significantly higher with respect to the 2VO+V group (Fig 5A and 5B).

**3.2.3. The low and high doses of lutein improved long-term synaptic plasticity.** The results indicated that the 2VO led to impairment in LTP induction of the Shaffer collateral-CA1synapses, but treatment with both low and high doses of lutein prevented this impairment (Fig 6A). The mean LTP induction for 60 min in the 2VO+V group (120.1 ± 2.3%) was significantly ($P < 0.001$) lower than the value of the sham group (169.9 ± 10.39%). Conversely, treatment with low and high doses of lutein significantly increases ($P < 0.01$) the LTP induction

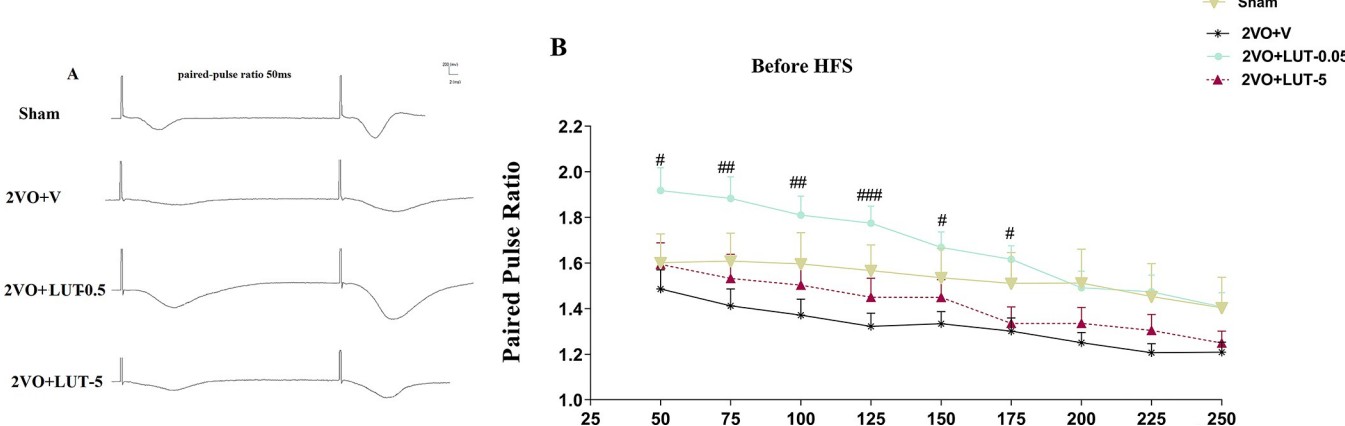

**Fig 5. The short-term synaptic plasticity is measured by paired-pulse ratio (PPR) calculation in ISIs 50 to 250 ms.** The low dose (0.5mg/kg) but not the high dose (5mg/kg) of lutein increased short-term synaptic plasticity in 2VO rats. Sample traces of responses (A). The linear graph for different ISIs (B). There was a significant difference between the 2VO + LUT-0.5 groups with respect to the 2VO + V group. The values are shown as mean ± SEM. The values are shown as mean ± SEM. Sham (n = 7), 2VO + V (n = 7), 2VO + LUT-0.5 (n = 11) and 2VO + LUT-5 (n = 8).

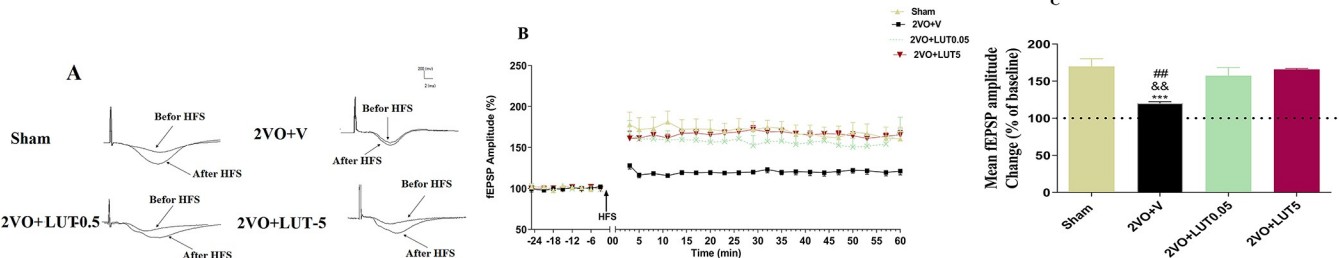

**Fig 6. Long-term synaptic plasticity (LTP) is induced by high-frequency stimulation (HFS) in different groups.** The low (0.5mg/kg) and high (5mg/kg) doses of lutein improved long-term synaptic plasticity. The sample traces from Schaffer collateral-CA1 synapses (A). The percentage change of the fEPSP amplitude compared to the baseline after high-frequency stimulation (HFS) (B). The comparison of mean-induced LTP after HFS between groups (C). Significant differences with respect to the sham (*** $P < 0.001$), 2VO + LUT-0.5 (## $P < 0.01$) and 2VO + LUT-5 (&& $P < 0.01$). The values are shown as mean ± SEM. Sham (n = 7), 2VO + V (n = 7), 2VO + LUT-0.5 (n = 11) and 2VO + LUT-5 (n = 8).

(157.4 ± 11.06% and 165.8 ± 1.05% respectively) with respect to the 2VO+V group and, the degree of LTP induction in the treated groups, comparable respect to the sham group. Furthermore, no significant difference was found between the magnitude of LTP in the 2VO+ LUT-0.5 and 2VO+ LUT-5 groups (Fig 6A–6C).

## 3.3. Treatment with a high dose of lutein restore the hippocampal cell loss and volume in the CA1 area

Fig 7 compares the hippocampal pyramidal cell number and volume respectively. The 2VO led to a 33.6% decrease (*P<0.001*) in the pyramidal cell population with respect to the sham group (331.6 ± 29.13 *vs* 500.1 ± 28.42 respectively). Although the treatment with low-dose lutein caused an increase in the number of cells (419.2 ± 21.12) and did not show any difference compared to the sham group, it was still not different from the 2VO+V group. In addition, the high-dose lutein administration also increased the number of pyramidal cells to 440.6 ± 15 value, which was comparable to the sham group and was higher than the value of the low-dose group (*P < 0.05*). Moreover, no significant difference was found between the pyramidal cell number of low-dose and high-dose lutein (Fig 7A and 7B).

A similar result was obtained in relation to the hippocampal volume so that this parameter in the 2VO+V group was 8.15 ± 0.287, which was statistically (*P < 0.01*) lower than that of the sham group (9.6 ± 0.341). While, the administration of low-dose lutein caused an increase in the hippocampal volume to 9.32 ± 0.312 value, but still did not show a statistical difference with respect to the 2VO+V group. Whereas the high dose of the lutein also increased the volume of the hippocampus, the results showed that the value of the hippocampal volume of the high dose (10.01 ± 0.363) treated group was not only comparable to the sham group but also statistically higher (*P < 0.01*) than the 2VO+V group (Fig 7C).

## 3.4. Treatment with a high dose of lutein improved the MDA level of 2VO rats

Fig 8 compares the MDA levels in different groups. The 2VO led to a significant (*P<0.01*) increase in the MDA level (4.58 ± 0.316) of the 2VO+V group with respect to the sham group (3.3673 ± 0.363). However, treatment with a low dose of lutein decreased MDA levels to 4.01 ± 0.289 but did not show a statistical difference compared to both sham and 2VO+V groups. Nevertheless, the high-dose lutein significantly decreased MDA levels (2.84 ± 0.183) with respect to the 2VO+V (*P<0.01*) and low-dose treated groups (*P<0.05*). Thus, the observed effect of high-dose lutein in the reduction of MDA levels exceeded that of low dose.

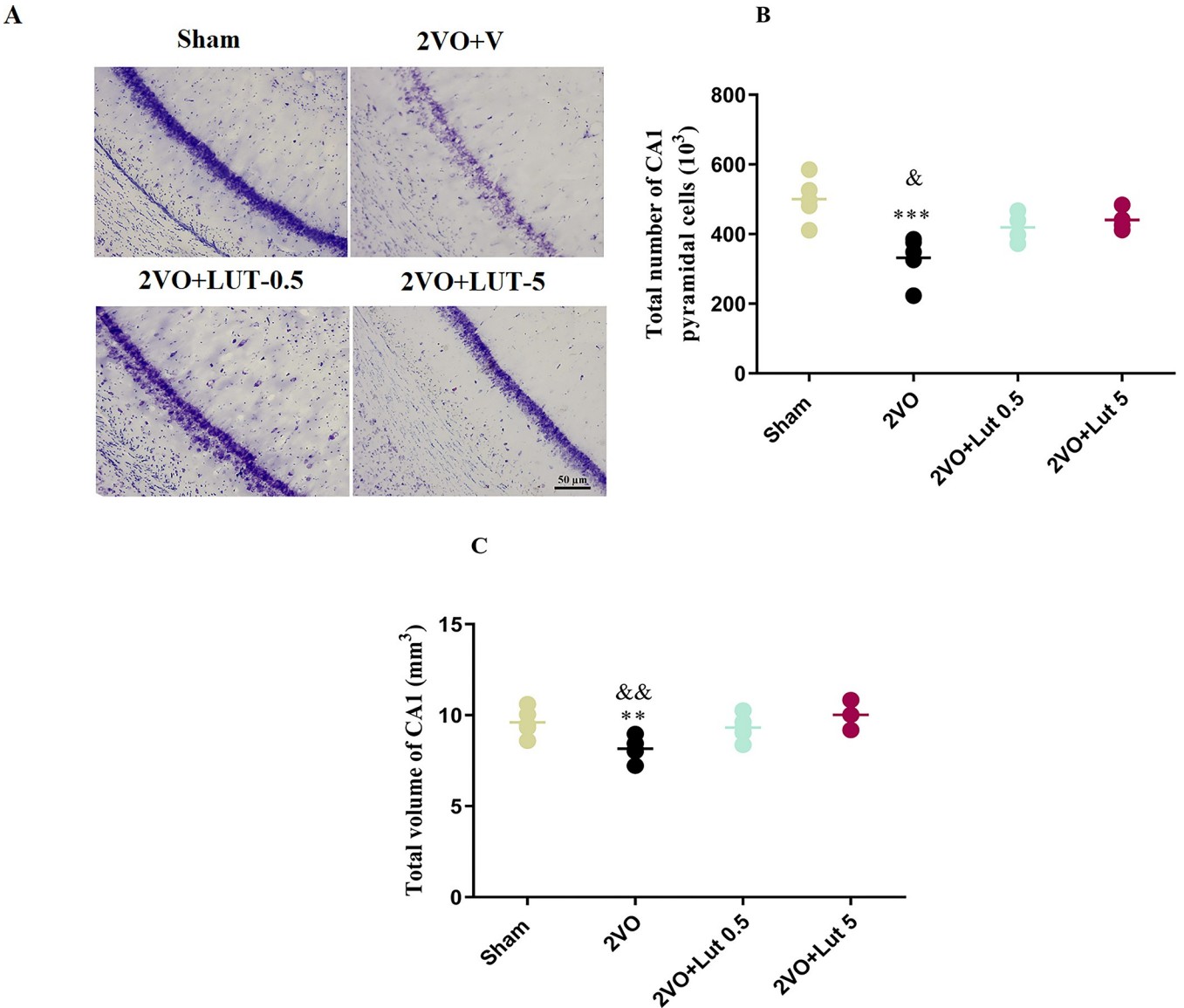

**Fig 7. Comparison of CA1 hippocampal pyramidal cell number and volume.** Treatment with a high dose (5mg/kg) but not a low dose (0.5mg/kg) of lutein restored the hippocampal cell loss and volume in 2-vessel occluded (2VO) rats. The image of light micro-photographs from CA1 pyramidal neurons (A). Total number (B) and volume (C) of hippocampal CA1 pyramidal cells. Significant differences with respect to the sham (* $P < 0.05$ and ** $P < 0.01$) and 2VO + LUT-5 (&$P < 0.05$ and && $P < 0.01$). The values are shown as mean ± SEM. Sham (n = 5), 2VO + V (n = 5), 2VO + LUT-0.5 (n = 5) and 2VO + LUT-5 (n = 5).

## 4. Discussion

The present study provides new finding that the treatment with a low and high dose of lutein improve spatial memory loss and long-term synaptic plasticity impairment in VD, but only administration of the high dose of lutein completely restores the fear memory, the hippocampal cell number and volume as well as MDA level. Interestingly, low dose, but not high dose, increased PPR with respect to the VD rats. Therefore, it seems likely that low doses and high doses of lutein improve memory and LTP through different mechanisms.

This study confirms that 2VO is associated with impairment in spatial and fear memory, which accords with our earlier observations [7, 8]. Treatment with high dose improved both

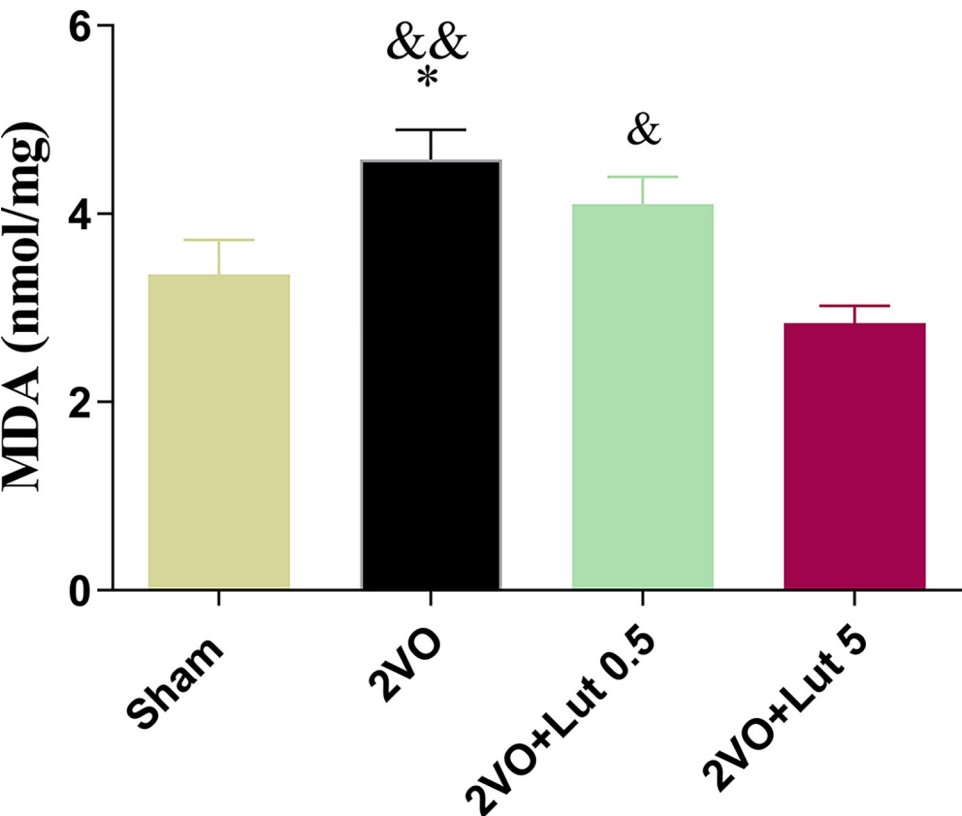

**Fig 8. Comparison of MDA levels of hippocampal between different groups.** Treatment with a high dose (5mg/kg) but not a low dose (0.5mg/kg) of lutein decreased the MDA level in 2-vessel occluded (2VO) rats. Significant differences with respect to the sham (* $P < 0.05$) and 2VO + LUT-5 (&$P < 0.05$ and && $P < 0.01$). The values are shown as mean ± SEM. Sham (n = 6), 2VO + V (n = 6), 2VO + LUT-0.5 (n = 6) and 2VO + LUT-5 (n = 6).

spatial and fear memory, but only spatial memory was restored and fear memory appeared to be unaffected by low dose administration. It's well known that spatial memory impairment in the MWM test could be due to damage to the hippocampus, but, fear memory impairment in passive avoidance tasks, could be due to damage to regions extending from the hippocampus to the amygdala and surrounding area [29]. These relationships may partly explain why the low-dose administration in this study failed to recover fear memory while the high dose restored both fear and spatial memory. It is therefore likely that treatment with high doses may affect several parts of the brain.

Consistent with the literature, this study found that memory impairment in VD rats is associated with a reduction in basal synaptic transmission (BST) and LTP [22]. The BST was evaluated by the I/O curve, and we found the partial recovery of basal neuronal excitability only in the high-dose lutein-treated group, and low-dose treatment did not show any effects on BST. In addition, the PPR, as an index for measurement of Glu release probability from presynaptic terminals, showed a non-significant but promising decrease trend in the 2VO group [30]. Surprisingly, treatment with low doses was found to significantly increase the levels of PPR in ISIs 50 ms to 175 ms with respect to the 2VO+V group. The I/O curve is an index for the evaluation of basal neuronal excitability, reflecting both the level of presynaptic neurotransmitter release and postsynaptic processes [31]. The depression of PPR indicates that the initial release probability was high. Conversely, the potentiation of PPR suggests that the initial release probability was low. Since, the 2VO+LUT-0.5 group displays an increase in PPR without the recovery of

BST, treatment with a low dose possibly led to a decrease in presynaptic neurotransmitter release probability. It has been reported that PPR facilitation, depends on two presynaptic factors including the immediately releasable vesicle pool and the probability of vesicular release [32]. It can therefore be assumed that the low dose of lutein might affect on release probability and/or releasable vesicles pool. It is not clear why the high dose did not affect PPR, a further study with more focus on PPR is therefore suggested. Despite the lack of changes in neurotransmitter release probability in the 2VO+LUT-5 group, the BST has been recovered, thus, treatment with a high dose (5mg/kg) perhaps affects postsynaptic processes, and treatment with the low dose (0.5 mg/kg) affects neurotransmitter release.

Another finding is that the MDA levels in the 2VO+V group significantly increased but high dose, not low dose administration of lutein restored the MDA level. The brain tissue is susceptible to oxidative damage because it has high lipid content, and low antioxidant capacity [33]. High levels of peripheral markers of oxidative stress and low antioxidant power have been reported in patients with MCI, late-onset AD, and VD. It is now understood that ROS increases neuronal damage and functional impairment in neurodegenerative diseases [34]. In accordance with the present results, a previous study also has demonstrated that lutein has a dose-dependent anti-inflammatory effect via inhibition of NF-κB [35]. The reduction of MDA with high-dose treatment can be a reason for the recovery of cell and volume number of the hippocampus, while the treatment with low dose probably failed to improve the volume and cell number of the hippocampus due to the lack of reduction of the MDA. Hence, increases in MDA could be an important factor, if not the only one, causing hippocampal neuron loss. Considering the improvement of memory and LTP, it seems likely that at least in the low-dose group, the remaining hippocampal neurons have improved functionally.

We also measured the LTP induction, as a specific cellular process for learning and memory. The result seems to be consistent with spatial memory, the VD memory-impaired animals exhibit deficits in LTP induction. This also accords with our earlier observations, which showed LTP impairment in 2VO rats [7, 8]. Interestingly, the LTP recovered by both low and high doses administration of lutein. The improvement of LTP in the high-dose treated group at least in part could be due to recovery of the volume and cell number of the hippocampus, MDA, as well as the BST. But, in the low-dose treated group, despite the LTP recovery, none of the above parameters had improved. Therefore, presynaptic neurotransmitter release probability could be at least in part an important reason for the improvement of LTP. There is general agreement that the pre-and post-synaptic mechanisms each individually or together can induce LTP in CA1 [36]. Thus, due to the improvement of LTP in the low-dose treated group, even without a change of BST, it is likely that pre-synaptic mechanisms are responsible for LTP recovery in this group. These results confirmed a presynaptic role for lutein in the low-dose group and it may cause the rescue of LTP in this group via presynaptic mechanisms.

## 5. Conclusion

Taken together, these results suggest that both low and high-dose administration of lutein show positive beneficial effects on spatial memory impairment and LTP induction in VD rats. However, high-dose administration of lutein decreases MDA levels and prevents cell loss and volume of the hippocampus associated with BST restoration, all these factors are known to play a role in LTP recovery and can be led to memory improvement in VD rats. Due to the lack of changes in PPR, LTP was restored at least in part through the post-synaptic mechanisms, which led to memory improvement in the high-dose treated group. Nevertheless, it is possible that the different mechanisms are responsible for spatial memory improvement and LTP recovery in low-dose received rats. Because the PPR was increased only in the low-dose

group, it seems that the low-dose administration of lutein rescue LTP induction through the neurotransmitter release probability and pre-synaptic mechanisms led to spatial memory improvement.

## Acknowledgments

We would like to express my sincere gratitude to Mrs. M. Mojahed for her invaluable technical assistance throughout the course of this project. Her expertise and dedication played a crucial role in enhancing the quality and precision of our work. We are truly appreciative of her contributions to the success of this endeavor.

## Author Contributions

**Conceptualization:** Amirhossein Yousefi Nejad, Somayeh Akbari, Seyed Mostafa Shid Moosavi, Masoud Haghani.

**Data curation:** Hamideh Asadi nejad, Amirhossein Yousefi Nejad, Somayeh Akbari, Maryam Naseh.

**Formal analysis:** Hamideh Asadi nejad, Somayeh Akbari, Maryam Naseh, Seyed Mostafa Shid Moosavi.

**Investigation:** Hamideh Asadi nejad, Amirhossein Yousefi Nejad, Somayeh Akbari, Maryam Naseh.

**Methodology:** Hamideh Asadi nejad, Amirhossein Yousefi Nejad, Somayeh Akbari, Maryam Naseh, Seyed Mostafa Shid Moosavi, Masoud Haghani.

**Project administration:** Masoud Haghani.

**Software:** Hamideh Asadi nejad, Amirhossein Yousefi Nejad, Somayeh Akbari, Maryam Naseh.

**Supervision:** Seyed Mostafa Shid Moosavi, Masoud Haghani.

**Validation:** Hamideh Asadi nejad, Somayeh Akbari, Maryam Naseh, Seyed Mostafa Shid Moosavi.

**Writing – original draft:** Hamideh Asadi nejad, Amirhossein Yousefi Nejad, Somayeh Akbari, Maryam Naseh, Seyed Mostafa Shid Moosavi, Masoud Haghani.

**Writing – review & editing:** Seyed Mostafa Shid Moosavi, Masoud Haghani.

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
