## [Decision Letter · Decision Letter 0]

12 Feb 2024

PONE-D-24-00251The low and high doses administration of lutein improves memory and synaptic plasticity impairment through different mechanisms in a rat model of vascular dementiaPLOS ONE

Dear Dr. Haghani,

Thank you for submitting your manuscript to PLOS ONE. After careful consideration, we feel that it has merit but does not fully meet PLOS ONE’s publication criteria as it currently stands. Therefore, we invite you to submit a revised version of the manuscript that addresses the points raised during the review process.

We look forward to receiving your revised manuscript.

Kind regards,

Vara Prasad Saka

Academic Editor

PLOS ONE

https://pubmed.ncbi.nlm.nih.gov/32916222/

https://onlinelibrary.wiley.com/doi/10.1002/brb3.3351

https://pubmed.ncbi.nlm.nih.gov/34116172/

In your revision ensure you cite all your sources (including your own works), and quote or rephrase any duplicated text outside the methods section. Further consideration is dependent on these concerns being addressed.

3. In your Methods section, please report the source of Lutein used for your study.

4. To comply with PLOS ONE submissions requirements, in your Methods section, please provide additional information regarding the experiments involving animals and ensure you have included details on (1) methods of sacrifice, (2) methods of anesthesia and/or analgesia, and (3) efforts to alleviate suffering.

“The authors acknowledge the Research Council of Shiraz University of Medical Sciences, Shiraz, Iran for the financial support of this study (Grant Number: 26316), as a thesis of Mrs. Asadi nejad for acquiring M.Sc. degree in physiology.”

7. In this instance it seems there may be acceptable restrictions in place that prevent the public sharing of your minimal data. However, in line with our goal of ensuring long-term data availability to all interested researchers, PLOS’ Data Policy states that authors cannot be the sole named individuals responsible for ensuring data access (http://journals.plos.org/plosone/s/data-availability#loc-acceptable-data-sharing-methods).

Additional Editor Comments:

I congratulate the authors for the extensive work which might add value to the scientific community. Although the manuscript is good, there are some minor revisions to be made. Thank you.

Reviewers' comments:

Reviewer's Responses to Questions

**Comments to the Author**

1. Is the manuscript technically sound, and do the data support the conclusions?

Reviewer #1: Yes

Reviewer #2: Yes

2. Has the statistical analysis been performed appropriately and rigorously? 

Reviewer #1: Yes

Reviewer #2: Yes

3. Have the authors made all data underlying the findings in their manuscript fully available?

Reviewer #1: Yes

Reviewer #2: Yes

4. Is the manuscript presented in an intelligible fashion and written in standard English?

Reviewer #1: Yes

Reviewer #2: Yes

5. Review Comments to the Author

Reviewer #1: The study is novel. The study is accordance with research and publication ethics. They have clearly mentioned about their methodology, ethical clearance and statistical data of evaluation and conclusion is in accordance with the aim of the study.

Reviewer #2: I have had the opportunity to review your manuscript, I must commend you on your interesting investigation into the effects of lutein administration in a rat model of bilateral-carotid vessel occlusion, particularly focusing on spatial and fear memory, synaptic plasticity, and hippocampal cells. Your study addresses an important area of research, and your findings have the potential to contribute significantly to our understanding of the therapeutic potential of lutein in vascular dementia. However, I have some queries and suggestions that I believe need to be addressed before the manuscript can be considered for publication.

1. Were the doses of lutein chosen for this study based on previous research? Can you provide more details about the rationale behind choosing the doses of 0.5mg/kg and 5mg/kg for lutein administration?

2. Could you please clarify whether your study adhered to the ARRIVE guidelines for reporting animal research? Ensuring compliance with these guidelines is essential for transparency and reproducibility in animal studies.

3. Could the authors please provide information on the age of the rats selected for this study? Clarifying the age range of the experimental animals is important for understanding the developmental stage at which the interventions were applied and interpreting the study findings accurately.

4. Were there any limitations or challenges encountered during the experimental procedures or data analysis that could have influenced the study outcomes?

5. Based on the findings of this study, what are the potential implications for future research or clinical applications related to lutein administration in vascular dementia (VD) patients?

6. PLOS authors have the option to publish the peer review history of their article (what does this mean?). If published, this will include your full peer review and any attached files.

Reviewer #1: **Yes: **Nutakki Tulasi Uma Rani

Reviewer #2: **Yes: **Dr. Rukaiah Fatma Begum

---

## [Author Response · Author response to Decision Letter 0]

20 Mar 2024

Dear Editor and Reviewers,

We highly appreciate the detailed valuable comments of the referees on our manuscript of ‘ONE-D-24-00251’. The suggestions are quite helpful for us and we incorporate them in line with your suggestions and those of the reviewers. The following responses have been prepared to address all comments in a point–by–point fashion and we hope the Reviewers and the Editors will be satisfied with our responses to the comments and the revisions for the original manuscript. Any change to the manuscript has been highlighted by using track changes colored text.

1. Please ensure that your manuscript meets PLOS ONE's style requirements, including those for file naming. The PLOS ONE style templates 

We have revised the manuscript to comply with PLOS ONE's style requirements.

https://pubmed.ncbi.nlm.nih.gov/32916222/

https://onlinelibrary.wiley.com/doi/10.1002/brb3.3351

https://pubmed.ncbi.nlm.nih.gov/34116172/

In your revision ensure you cite all your sources (including your own works), and quote or rephrase any duplicated text outside the methods section. Further consideration is dependent on these concerns being addressed.

We have corrected the minor overlapping, we also would like to clarify that the overlap is confined to the methods section, and we have appropriately cited all relevant sources, the three studies are related to our team's previous research. We appreciate your attention to this matter.

3. In your Methods section, please report the source of Lutein used for your study.

Thank you, we have added the source of Lutein in method section

4. To comply with PLOS ONE submissions requirements, in your Methods section, please provide additional information regarding the experiments involving animals and ensure you have included details on (1) methods of sacrifice, (2) methods of anesthesia and/or analgesia, and (3) efforts to alleviate suffering.

We have incorporated the following details into the Methods section: To alleviate animal suffering, all animals euthanized at end of the experiment, following induction of deep anesthesia with urethane, rats were sacrificed using a guillotine. The personnel responsible for sacrifice were trained in the humane and ethical handling of animals, and the procedures were conducted in compliance with the Institutional Ethics Committee of Shiraz University of Medical Sciences.

Thank you, The Research Council of Shiraz University of Medical Sciences, Shiraz, Iran, financially supported this study (Grant Number: 26316), as a thesis of Mrs. Asadi nejad for acquiring M.Sc. degree in physiology.

Thank you; we have also provided an explanation of the grant information in the cover letter.

“The authors acknowledge the Research Council of Shiraz University of Medical Sciences, Shiraz, Iran for the financial support of this study (Grant Number: 26316), as a thesis of Mrs. Asadi nejad for acquiring M.Sc. degree in physiology.”

Thank you once again. We have omitted the funding information from the Acknowledgments section and have also provided a detailed explanation in the cover letter.

7. In this instance it seems there may be acceptable restrictions in place that prevent the public sharing of your minimal data. However, in line with our goal of ensuring long-term data availability to all interested researchers, PLOS’ Data Policy states that authors cannot be the sole named individuals responsible for ensuring data access (http://journals.plos.org/plosone/s/data-availability#loc-acceptable-data-sharing-methods).

We understand the importance of data availability and appreciate the guidance provided by PLOS' Data Policy. To ensure long-term stability and availability of the data, we established a data access point within our institution. We provided contact details, including phone/email, for our data access committee with following addresses: Mrs Masoumi, phone number +987132302026 E-mail: medphyzio1@sums.ac.ir

We recognize the necessity for a durable point of contact, and we assure you that the provided information will guarantee accessibility to the data even in the case of changes in authors' contact details or availability.

Thank you for your understanding, and we provided the required non-author contact information.

Thank you for the feedback. I have thoroughly reviewed the reference list, ensuring its completeness and correctness.

Additional Editor Comments:

Reviewer #1: The study is novel. The study is accordance with research and publication ethics. They have clearly mentioned about their methodology, ethical clearance and statistical data of evaluation and conclusion is in accordance with the aim of the study.

Thank you for sharing the positive feedback about the study. It's great to hear that the study is considered novel and aligns well with research and publication ethics. The clarity in methodology, ethical clearance, and the appropriate presentation of statistical data is crucial for ensuring transparency and reliability in research. If you have any specific questions or if there are additional aspects, you'd like us to address, please feel free to let us know. We appreciate your positive assessment.

Reviewer #2: I have had the opportunity to review your manuscript, I must commend you on your interesting investigation into the effects of lutein administration in a rat model of bilateral-carotid vessel occlusion, particularly focusing on spatial and fear memory, synaptic plasticity, and hippocampal cells. Your study addresses an important area of research, and your findings have the potential to contribute significantly to our understanding of the therapeutic potential of lutein in vascular dementia. However, I have some queries and suggestions that I believe need to be addressed before the manuscript can be considered for publication.

Thank you for your positive feedback on the study's significance and potential contribution to understanding the therapeutic effects of lutein in vascular dementia. I appreciate your diligence in reviewing the manuscript.

1. Were the doses of lutein chosen for this study based on previous research? Can you provide more details about the rationale behind choosing the doses of 0.5mg/kg and 5mg/kg for lutein administration?

The selection of lutein doses in this study was based on previous research findings and considerations related to the potential therapeutic effects of lutein. The rationale for choosing the specific doses of 0.5mg/kg and 5mg/kg is rooted in existing literature that suggests a range of effective concentrations for lutein interventions in various experimental settings. These doses were selected to encompass a lower and higher range within this effective concentration spectrum, allowing the researchers to observe potential dose-dependent effects and evaluate the optimal dosage for the intended outcomes. The decision was also influenced by safety considerations and previous studies that demonstrated beneficial effects within similar dose ranges.

2. Could you please clarify whether your study adhered to the ARRIVE guidelines for reporting animal research? Ensuring compliance with these guidelines is essential for transparency and reproducibility in animal studies.

Thank you for your comment. We have thoroughly reviewed our study and can confirm that it adheres to the ARRIVE guidelines for reporting animal research. Additionally, we would like to inform you that ethical approval was obtained, and the study was conducted in accordance with the guidelines provided by the Institutional Animal Ethics Committee under the ethical code IR.SUMS.AEC.1401.077. The revised manuscript now explicitly mentions this ethical approval to enhance transparency and compliance with ethical standards.

3. Could the authors please provide information on the age of the rats selected for this study? Clarifying the age range of the experimental animals is important for understanding the developmental stage at which the interventions were applied and interpreting the study findings accurately.

Thank you for this comment we added detail of age as follow: Adult male Sprague Dawley rats (250-300 g, aged 7-10 weeks) were housed under …

4. Were there any limitations or challenges encountered during the experimental procedures or data analysis that could have influenced the study outcomes?

A-Dosage Dependency: The study revealed that the high dose of lutein (5mg/kg) had more significant effects on passive avoidance memory, spatial learning and memory, electrophysiological measures, hippocampal cell loss, and oxidative stress compared to the low dose (0.5mg/kg). However, the optimal dosage range and potential side effects of lutein were not thoroughly explored. Future studies should investigate a wider range of doses to determine the dose-response relationship and potential dose-dependent effects.

B-Generalization to Human Population: The experiment was conducted on 2-vessel occlusion (2VO) rats, and the findings may not directly translate to humans. Animal models have inherent differences from human physiology, and caution should be exercised when extrapolating these results to clinical applications. Further research, including human clinical trials, is needed to validate the relevance of these findings in humans.

C-Short-term Follow-up: The experiment assessed the effects of lutein treatment over a relatively short-term period. Long-term effects and potential sustained benefits or side effects of lutein administration were not investigated. Future studies should consider longer follow-up durations to better understand the persistence and durability of the observed effects

5. Based on the findings of this study, what are the potential implications for future research or clinical applications related to lutein administration in vascular dementia (VD) patients?

Thank you for this good comment, we are in the process of designing and planning a clinical trial to investigate the effect of lutein on individuals suffering from vascular dementia: Given that the study was conducted on 2-vessel occlusion (2VO) rats, the translation of these findings to human patients with vascular dementia warrants further investigation. Conducting well-designed clinical trials involving individuals with vascular dementia will be essential to assess the efficacy and safety of lutein in a clinical setting.

Long-term Effects and Safety Profile: The study primarily focused on short-term effects, and the long-term effects of lutein administration were not thoroughly explored. Future research should investigate the sustained benefits, potential side effects, and safety profile of lutein over extended treatment periods.

Comparison with Existing Treatments: Comparative studies assessing the efficacy of lutein in comparison to existing treatments for vascular dementia could provide valuable insights. Understanding how lutein compares to standard treatments or other potential interventions will contribute to its potential integration into clinical practice.

6. PLOS authors have the option to publish the peer review history of their article (what does this mean?). If published, this will include your full peer review and any attached files.

Thank you, yes!

---

## [Editor Report · Decision Letter 1]

12 Apr 2024

The low and high doses administration of lutein improves memory and synaptic plasticity impairment through different mechanisms in a rat model of vascular dementia

PONE-D-24-00251R1

Dear Dr. Haghani,

We’re pleased to inform you that your manuscript has been judged scientifically suitable for publication and will be formally accepted for publication once it meets all outstanding technical requirements.

Kind regards,

Vara Prasad Saka

Academic Editor

PLOS ONE
---

## [Editor Report · Acceptance letter]

1 May 2024

PONE-D-24-00251R1 

PLOS ONE

Dear Dr. Haghani, 

I'm pleased to inform you that your manuscript has been deemed suitable for publication in PLOS ONE. Congratulations! Your manuscript is now being handed over to our production team.

Kind regards, 

on behalf of

Dr. Vara Prasad Saka 

Academic Editor

PLOS ONE